# A Review of Selected Studies on the Factors Associated with the Nutrition Status of Children Under the Age of Five Years in South Africa

**DOI:** 10.3390/ijerph17217973

**Published:** 2020-10-30

**Authors:** Mbalenhle Mkhize, Melusi Sibanda

**Affiliations:** Department of Agriculture, University of Zululand, Private Bag X1001, KwaDlangezwa 3886, South Africa; mbalimkyeze@gmail.com

**Keywords:** food insecurity, malnutrition, micro-nutrient deficiency, stunting, under-nutrition, wasting

## Abstract

Malnutrition is a considerable contributor to child mortality and morbidity. Child malnutrition further affects the country’s economic development. Child malnutrition in South Africa is persistent, continuing to be an alarming burden. The nutritional status of kids under the age of five years is a critical indicator of the country’s economic condition and health status. An understanding of the influencers of the nutritional status of children can act as a catalyst in combatting all forms of malnutrition. The purpose of this paper was to review selected studies concerning the factors that affect the nutritional status of children in South Africa. Studies were selected from electronic databases, which were PubMed, Google Scholar, Science Direct, Sabinet African Journals, and the University of Zululand library catalog. The keywords that were used to search studies and articles from the selected database were: risk factors, child nutritional status, children under the age of five years, South Africa, malnutrition, underweight, stunted, wasting, and over-nutrition. Studies and surveys published from 2010–2019 that reported on the factors influencing the nutritional status of children under the age of five years were included in this review. Twenty-seven articles met the inclusion criteria of the study. The 27 articles were made up of 21 cross-sectional articles and six longitudinal articles. The finding from this review highlights that there is a lack of studies conducted in urban areas. The results show that the nutritional status of children is affected by several factors. These include household food insecurity, low household income, illiterate caregivers, unemployment, inadequate dietary intake, low birth weight, consumption of monotonous diets, poor caregiver’s nutritional knowledge, poor access to water and sanitation, poor weaning practices, age of the caregiver, and demographic characteristics of a child (age and gender). It is critical to have an understanding of the factors that affect the nutritional status of children. Such knowledge can significantly contribute to formulating policies that can enhance nutrition security and the country’s economy. Moreover, insights into strategic interventions to eradicate all forms of malnutrition can be made.

## 1. Introduction

Malnutrition is regarded as an intensifying global health challenge that is linked to high-cost care, illness, and death [1]. Childhood malnutrition may also result in long-term effects that are irreversible, such as delayed cognitive and physical development [2]. Malnutrition further diminishes sensory-motor abilities, reproductive capacity, and makes the child more subjected to hereditary diseases, such as diabetes, reducing productivity in working capital at adulthood [3].

The problem of child malnutrition is highly concentrated in low and middle income countries, whereby at least one-third of children are malnourished [4,5]. South Africa has come a long way in eradicating poverty and malnutrition [6]. Despite the economic advancement post-apartheid, South Africa has been experiencing a drastic nutrition transition, which has led to malnutrition and other health challenges [7]. The drastic nutrition transition in South Africa has been due to the depreciation of the economy, which has resulted in the high poverty rate. The high poverty rate has caused a dietary shift amongst South Africans as a strategy to mitigate hunger [8]. In 2018, an estimated 59% of children lived below the poverty line [9]. Children who live below the poverty line are nutritionally deprived and malnourished [10].

The nutritional status of children under the age of five years is regarded as a proxy of the country’s economic condition [11]. Therefore, low nutritional status in children is not only a public health challenge, but it also affects the country’s economic situation. In South Africa, an estimated 1.4% loss in economic productivity is due to childhood stunting. About 1.1 billion is lost every year in gross domestic product (GDP) due to micronutrient deficiency arising from child malnutrition [11]. South Africa is amongst the top 20 countries globally with the uppermost burden of under-nutrition. It is further classified as one of the 36 countries with a high prevalence of malnutrition [12].

In South Africa, there is documented evidence of the triple burden of malnutrition, which includes under-nutrition, over-nutrition, and micronutrient deficiency [13,14]. Stunted growth is the most prevalent form of malnutrition in South Africa, accounting for 80% of the world’s stunted children [15]. In 2016, an estimated 27% of children under the age of five years in South Africa were stunted, 44% had a vitamin deficiency, 13% were overweight, and 6% were underweight [13]. Incidents of malnutrition in South Africa differ across geographical areas and socioeconomic groups. In 2017, Gauteng was the leading province with the highest proportion (34.20%) of children who were stunted, followed by Free State (33.50%), KwaZulu-Natal (28.50%), North West (12.60%), and Western Cape (11.90%) [14].

Despite all programs and policies set to combat hunger, malnutrition is still persistent in South Africa. The influencers of the nutritional status in children under the age of five years are not well understood in South Africa. Available studies are geographically limited primarily to rural areas. Yet, poverty-stricken urban areas have shown a high prevalence of child malnutrition.

Global scholars convey several factors that cause child malnutrition. Factors that cause child malnutrition differ with geographical location. Child malnutrition in low and middle income countries is also highly manifested by overweightness and obesity, which result from the consumption of food with high calorie content without dispensing the energy consumed through physical activities [16,17]. Studies conducted in low and middle income countries show similar factors that cause child malnutrition, factors such as household food insecurity, low household income, lack of access to adequate clean water, insufficient health facilities, low educational level, distance to health facilities, hygiene, and poor sanitation [18,19,20,21]. The Action Against Hunger [22] identified poverty as the root cause of malnutrition that deprives the attainment of sufficient nutrients from the consumed foods amongst the vulnerable population. Similarly, Bernstein [23] stated that poverty increases the risk of an individual to be affected by infectious diseases, which ultimately cause malnutrition.

Little is known about specific factors influencing the nutritional status of children under the age of five years at a micro level in different geographical locations in South Africa. This situation triggers a need to generate knowledge about factors influencing the nutritional status of children under the age of five years in South Africa. Therefore, this review paper intends to fill this research gap by assessing the factors that influence the nutritional status of children under five years in South Africa using available literature. The United Nations International Children Emergency Funds’ theoretical framework of causes of child malnutrition groups factors that cause child malnutrition into three groups, which are basic, underlying, and immediate causes [24]. The UNICEF theoretical framework of causes of child malnutrition can act as a catalyst in achieving the objective of this review, by providing guidelines on the existing theory about the causes of child malnutrition. The anticipation is that the findings can significantly contribute to generating information that could be useful in formulating policies that can enhance the nutrition security among children in South Africa. Moreover, insights into strategic interventions to eradicate all forms of malnutrition can be made. Systematic literature reviews and meta-analyses usually follow the Preferred Reporting Items for Systematic Reviews and Meta-Analyses (PRISMA), with evidence-based minimum set criteria and interventions. This paper does not claim to be a systematic review nor a meta-analysis. We also do not seek to make interventions but rather a synthesis of the literature to understand the situation concerning the nutrition status of children under five years of age in South Africa. However, to make the literature review coherent, we structured the review paper to include some background information, an objective, a description of data sources and eligibility criteria, results and discussion (implications), limitations, and conclusions.

## 2. Materials and Methods

This paper reviewed the literature on factors associated with the nutrition status of children under five years of age in South Africa from 2010 to 2019. This paper extracted relevant information from literature sources. To ensure that the extracted data was descriptive, we applied a robust selection and exclusion criteria when selecting studies. The inclusion and eligibility criteria are explained in the following section.

### 2.1. The Inclusion and Eligibility Criteria

Longitudinal and cross-sectional studies that determined factors that influence the nutritional status of children under five years of age were considered in this review. This review focused on studies undertaken only in South Africa. The review fixated on peer-reviewed papers published in international scientific journals. Only high-quality articles were considered. High-quality articles were those that met the inclusion criteria for this review and had a meaningful sample size of at least 100 children under the age of five years. This meant that grey literature, such as unpublished studies, conference proceedings, and those that were not peer-reviewed were not included. Although the risk of bias assessment was not a strict requirement in evidence synthesis methods outside of systematic reviews, to avoid bias, we identified all available studies on the topic for the South African context. We also applied an eligibility criterion as described in the inclusion/exclusion criteria (Table 1). Moreover, we incorporated and reported the findings in the review, and interpreted them as supported by empirical evidence on the topic from other sources. Reviews and conceptual and explorative articles were excluded to ensure that primary data was obtained. The literature search was limited to studies published in English.

Over the past decade, urbanization, political, economic, and social-demographic transition have been witnessed in South Africa [25]. Therefore, this review only included recent studies undertaken from 2010 onwards, and this was done to reflect the contemporary literature that is linked to the transitions that have been taking place in South Africa. Table 1 summarizes the inclusion and eligibility criteria for this review.

### 2.2. Search Strategy

Articles were searched in the following databases: PubMed, Google Scholar, Science Direct, Sabinet African Journals, and the University of Zululand library catalog search for journals. Key terms that were used to search studies from the selected databases were: risk factors, child nutrition status, children under five years, South Africa, malnutrition, underweight, stunted, wasting, and over-nutrition. In addition to the key terms, factors influencing the nutritional status of children under the age of five years were hypothesized based on the theoretical framework of causes of child malnutrition provided by the United Nations International Children’s Emergency Fund (UNICEF).

Factors included basic factors (human, economic, and organizational resources), underlying factors (such as household food insecurity, illiteracy, insufficient maternal and child care, inadequate health services, and poor environmental conditions), and immediate factors (such as diseases and inadequate dietary intake). Figure 1 illustrates the causes of malnutrition per UNICEF’s theoretical framework.

Neighbouring countries such as Botswana and Namibia identified low birth weight as one of the contributing factors to child malnutrition [26,27]. Therefore, low birth weight was also hypothesized as a factor that may influence the nutritional status of children under the age of five years. Articles were initially screened to ensure that they were not duplicated. Abstracts and titles were read, and articles were selected per the set criteria (Table 1). Articles with no abstract and full text were read and also screened based on the set criteria. After all of the screening was done, 27 studies met the criteria for inclusion and were selected for this review for a detailed analysis. The article search commenced on 21 November 2019, and it was completed on 9 January 2020. Figure 2 shows the flow chart for the study selection process.

### 2.3. Data Extraction and Analysis

Relevant data were extracted and further analyzed in terms of the characteristics of the included articles. Data were separated in terms of the region covered, such as provinces, urban or rural areas, and informal settlements. Information based on the study date of collection, study design, setting, sample size, nutritional status, and factors influencing nutritional status were extracted independently. Studies with data that were not clear or contradicting and information obtained through reading the discussions of the articles were excluded in this paper.

## 3. Results

### 3.1. Description of the Methodological Design of Selected Studies

A majority (77.78%) of the reviewed studies were cross-sectional (Figure 3). The remaining 22.22% of the reviewed studies were longitudinal, and the duration for the longitudinal studies was two years. The number of children recruited (range) in the review studies was from 166 to 500 children.

### 3.2. Geographical Distribution of the Reviewed Studies by Province

A higher proportion (40% and 29%) of the studies were confined to the Limpopo and Gauteng Provinces, respectively (Table 2). There were 22.22% of the reviewed studies that focused on the Eastern Cape and KwaZulu-Natal Provinces each (Table 2). About 15% of the studies were conducted in both Mpumalanga and Western Cape (Table 2). The remaining 11% of the studies were concentrated in the Northern Cape, Free State, and North West Provinces each. Some studies were conducted in more than one province (Table 2). Table 2 summarizes the geographical distribution of the reviewed studies.

### 3.3. Geographical Classification of the Reviewed Studies (Rural or Urban)

The majority (62.96%) of the reviewed studies were primarily conducted in rural areas. Of the remaining 33.30% of the reviewed studies that were conducted in urban areas, 14.81% and 7.41% were in informal urban settlements and peri-urban areas, respectively.

### 3.4. The Evolution of the Number of Studies on Child Nutritional Status in South Africa

Between 2010–2014, only 44.44% of the reviewed studies focused on the factors influencing the nutritional status of children under five years in South Africa. There was a progressive increase (55.56%) in the number of studies undertaken between 2015–2019. Figure 3 illustrates the number of studies conducted on the factors influencing the nutritional status of children under five years in South Africa from 2010–2019.

### 3.5. The Prevalence of Child Malnutrition in South Africa as Depicted by the Selected Studies

The prevalence of malnutrition was estimated for each province using the nutritional indicators. The average prevalence percentage of each nutritional indicator was calculated for each province from the selected studies. Studies conducted in the Limpopo Province showed a high (69.90%) prevalence of underweightness (Figure 4). Studies conducted in the Gauteng Province reported stunting as the highest form of malnutrition within the province, responsible for 11.88% (Figure 4), while studies conducted in the Eastern Cape Province showed that stunting had the highest (14.46%) prevalence (Figure 4). Studies conducted in the KwaZulu-Natal Province showed a high (15.25%) prevalence of stunting. In the Mpumalanga Province, stunting growth accounted for 5.28% (Figure 4). In the Free State Province, stunting growth accounted for 7.30% (Figure 4). Studies conducted in the North West province reflected the prevalence of stunting to be 16.83% (Figure 4). Overweightness/obesity was high (10.23%) in the Western Cape Province (Figure 4). Finally, studies conducted in the Northern Cape Province reflected a high (22.20%) prevalence of wasting within the province (Figure 4).

### 3.6. Nutritional Status Indicators in Children

Stunting was the highest (85.18%) nutritional indicator in children that represented malnutrition status from the reviewed studies (Figure 5). Stunting was followed by underweightness, which accounted for 62.96%, wasting (51.85%), overweightness/obesity (25.92%), and micronutrient deficiency (18.51%) (Figure 5). Figure 5 shows the nutritional status indicators of children from the reviewed studies.

### 3.7. Factors Influencing the Nutritional Status of Children

The reviewed studies depicted several factors that influence the nutritional status of children under the age of five years in South Africa. Nearly half (48.14%) of the reviewed studies identified household food insecurity as the primary factor that affected the nutritional status of children under the age of five years (Figure 6). Low household income was also represented by a higher proportion (40.70%) of the reviewed studies as a factor that influenced the nutritional status of children under the age of five years (Figure 6). Other factors included the caregiver’s level of education, reported by 37.03% of the studies, unemployment (29.62%), and inadequate dietary intake, low birth weight, and child illness were each reported by 25.92% of the selected studies (Figure 6). Other factors that were identified were: monotonous diet, poor caregiver’s nutrition education, poor access to water and sanitation, poor weaning practices, the gender of the child, age of the child, and that of the caregiver (Figure 6). Figure 6 summarizes the different factors that influenced the nutritional status of children identified by the reviewed studies under five years of age in South Africa.

Table 3 summarises the findings of this review on the factors that influenced the nutritional status of children under five years of age in South Africa using data from the selected studies.

## 4. Discussion

### 4.1. Geographical Classification of the Reviewed Studies

This study aimed to review selected studies conducted in South Africa concerning the factors that affect the nutritional status of children under the age of five years. The finding from this review shows that studies conducted on the factors that influence the nutritional status of children are not evenly distributed across the provinces in South Africa. Most studies were concentrated in the Limpopo Province. Most studies were conducted in Limpopo because it is mainly dominated by rural areas, which are perceived to have high levels of poverty and malnutrition. This finding suggests that the factors affecting the child nutritional status are not well documented across all provinces in South Africa. As a result, there may be inadequate policy and strategic intervention to eradicate malnutrition in other provinces. Overall, results from this review show that a high proportion of studies were conducted in rural areas, while few were conducted in urban areas. This implies that there is still a growing census of perceptions that poverty and malnutrition are mainly found in rural areas. Recent studies show that the locus of poverty is shifting from rural areas to urban areas [52]. This finding is supported by Zhou et al. [53]. They stated that the rapid growth of urbanization has resulted in poverty and nutrition transition, which has ultimately caused the burden of malnutrition. As more people migrate from rural areas to urban areas for better lives, urban areas became overpopulated, creating the competition for livelihood resources [54].

Moreover, the rapid growth of urbanisation increases the number of households who live in informal settlements [55]. Therefore, the rapid growth of informal settlements around the world poses a threat to the world’s nutritional status. Individuals who live in informal settlements are subjected to health challenges throughout a lifetime [56]. Therefore, children living in informal settlements are most likely to be vulnerable to malnutrition.

### 4.2. Evolution of the Number of Studies on Child Nutritional Status in South Africa

The review shows that between 2010–2014 and 2015–2019, the number of studies has increased by 44.4% and 55%, respectively. This drastic increase may be due to the need to tackle the persistent problem of child malnutrition in South Africa that can be linked to the economic slowdown South Africa has been experiencing over the past years. The economic downturn does not neglect the nutritional status of children [57]. The growth of studies on the factors influencing the nutritional status over the years shows that the nutritional status of children and child malnutrition is a persistent health challenge. The majority of the reviewed studies selected for this paper were cross-sectional and a few longitudinal.

Longitudinal studies can be beneficiary in identifying historical trends to identify possible solutions and recommendations [58,59]. Globally, scholars have proven that child malnutrition can be clinically manifested at the later stage of life [60]. Therefore, longitudinal studies are recommended to identify the growth development of the child [61]. This pattern highlights the need for more longitudinal studies in South Africa that will provide a more representative trend of child malnutrition. The range that describes the number of children in studies reviewed and the duration of the follow up for the longitudinal studies is two years.

### 4.3. The Prevalence of Child Malnutrition in South Africa

Findings from this review reflect a high prevalence of malnutrition across different provinces, where stunting growth is found to be the leading form of malnutrition affecting children under the age of five years. These findings are in line with the South African national survey conducted by Ilifa Labantwana [13] showing that stunting is the most prevalent form of malnutrition in South Africa. Similarly, Chakona and Shackleton [12] stated that South Africa is amongst the top 20 countries globally with the uppermost burden of under-nutrition, and WHO further classifies it as one of the 36 countries with a high prevalence of malnutrition manifested by stunting. As alarming as the prevalence of malnutrition among children in South Africa is, it is not severe compared to other low and middle income countries across the globe. Studies conducted in middle income countries have shown a high prevalence of child malnutrition [62]. For example, Akombi et al. [5] showed that at least one-third of children in Sub-Sahara Africa are malnourished. This finding is supported by a report from the Centre for Disease Control and Prevention [63] that revealed that one-third of children globally are suffering from micronutrient deficiency. Sub-Sahara Africa is leading by 48% in micronutrient deficiency in children, followed by South Asia with 44%. A study conducted in an urban slum of Pune, Maharashtra, India revealed that child malnutrition is also an alarming challenge amongst Indian society, affecting one in every five children [64]. This implies that child malnutrition is a global challenge, not only prevalent in South Africa, that needs urgent attention.

#### 4.3.1. The Prevalence of Child Stunting in South Africa

Stunting was the most prevalent form of malnutrition present in South Africa reported by the reviewed studies. This finding is a concern for public health and to the economic advancement of South Africa. Stunting has been identified as the primary indicator that has hindered the achievement of the first Millennium Development Goals (MDGs) [65]. Besides, stunting is also a health challenge that prevents children under the age of five years from achieving their childhood milestones [42,45]. Kimani-Murage et al. [66] indicated that for every five children in South Africa, one is stunted. A national survey conducted in South Africa by Ilifa Labantwana [13] shows that an estimated 31% of children under the age of two years are stunted. Stunting is also a global concern. For example, a study conducted in Sub-Saharan African countries, excluding South Africa, reported a high (28%) prevalence of stunting in children under the age of five [67]. Moreover, stunting is highly seen in countries situated in South Asia, responsible for 38% of stunted children [68].

#### 4.3.2. The Prevalence of Underweight Children in South Africa

Underweightness has been classified as the nutritional indicator that can be used to evaluate and monitor the nutritional status of children [69,70]. Despite the growing number of studies showing the increase in the number of underweight children, the situation of childhood underweightness in South Africa is not as severe as other low and middle income countries. Studies globally reveal underweightness as a nutritional indicator that is more of an epidemic in low and middle income countries such as West Africa, Eastern Africa, and South Asia [71,72,73].

#### 4.3.3. The Prevalence of Child Wasting in South Africa

According to UNICEF [74], wasting is more severe in emergency settings, with high levels of hunger and communicable diseases. The United Nations International Children’s Emergency Fund [74] further reported that an estimated 49.50 million children under five years are wasting globally, out of which at least 15.20% are found in South Asia, followed by countries in West and Central Africa (9%). Results from this review show that about half of the reviewed studies identified wasting as a nutritional indicator that reflects childhood malnutrition in South Africa. This finding suggests that wasting in South Africa is persistent and continues to be an alarming health challenge. Scholars globally agree that wasting is mainly attributed to hunger and frequent child illness [75,76,77].

The World Health Organisation [78] identified inadequate healthcare services, improper feeding practices, and poor sanitary environment as significant contributors to child wasting. According to UNICEF [74], a high (76%) proportion of children in Africa are confronted with hunger levels that exacerbate vulnerability to wasting. Furthermore, the increase in areas that are classified as poor hygiene areas and low service delivery (informal settlements) further increases the risk of child wasting in South Africa [79].

#### 4.3.4. The Prevalence of Overweightness and Obese Children in South Africa

The nutritional status of children is also reflected by overweightness and obesity [78]. However, some scholars classify overweightness and obesity as a condition that can be genetically inherited [80,81]. Overweightness and obesity are regarded as a nutritional indicator that results from excessive intake of nutrients compared to the quantity needed by the body and without engaging in physical activities [82]. Overweightness and obesity are triggered by the consumption of foods that are high in unsaturated fats and beverages that have high sugar content [83,84]. Cohen et al. [85] classify the nutrition transition that may have resulted from urbanization as the major cause of overweightness and obesity. Historically, overweightness and obesity were mainly a health challenge that affected wealthy countries [74]. However, recent studies have validated overweightness and obesity even in low and middle income countries, including South Africa [86,87,88]. Findings from this review show that some selected studies reported overweightness and obesity as a nutritional indicator that is an epidemic in South Africa. This finding may be due to rapid urbanisation, which has led to the nutrition transition.

Mtolo [89] validated that overweightness and obesity in South Africa are due to the economic and nutrition transition. Global scholars show that overweightness and obesity is a growing concern for children under the age of five years. The United Nations International Children’s Emergency Fund [74] reported that there is an increasing number of children who are overweight or obese globally, whereby an estimated 40 million children are overweight or obese.

#### 4.3.5. The Prevalence of Micronutrient Deficiency of Children in South Africa

Micronutrient deficiency is also used as a nutritional indicator to describe the nutritional status of a child. Results from this review show that few studies identified micronutrient deficiency as a nutritional indicator that reflected the nutritional status of children in South Africa. However, a national survey conducted by the Ilifa Labantwana [13] validated that micronutrient deficiency in South Africa is an epidemic, whereby vitamin deficiency is highly observed among South African children. Global scholars reflect micronutrient deficiency as an epidemic form of malnutrition that results from an imbalanced diet [90,91]. According to the Centre for Disease Control and Prevention [63], half of children under the age of five years globally suffer from at least one or more forms of micronutrient deficiency. The health effects of micronutrient deficiency include weight loss, immune dysfunctioning, impaired physical growth, neurological disorders, cardiovascular diseases, megaloblastic anaemia, and skin problems [90]. Micronutrition deficiency in children is shown by several factors, such as numbness, jaundice, swollen tongue, anaemia, weakness, and fatigue [91]. The small number of studies in South Africa that identify micronutrient deficiency as a nutritional indicator that reflects the nutritional status of children suggests a need for studies that will assess child malnutrition using micronutrient deficiency as a proxy. The growing number of studies that show the presence of various nutritional indicators suggests the need for conducting a survey that provides precise knowledge of the factors that influence the nutritional status of children. As explained in the previous section, the nutritional status of children under the age of five years is a critical indicator of the country’s economic condition and health status [92]. It is, therefore, vital to assess and understand the factors that influence the nutritional status of children to develop strategies to combat all forms of malnutrition and enhance the country’s economic growth. This review revealed several common factors that influence the nutritional status of children.

### 4.4. Factors Affecting the Nutritional Status of Children in South Africa

#### 4.4.1. Food Insecurity

Food insecurity was identified as the principal factor that affects nutritional status in South Africa. This finding implies that despite all strategies developed to reduce food insecurity in South Africa, such as the food integrated nutrition programme, zero hunger provision of social grants, and food parcels, food insecurity continues to be a challenge. According to Madiba et al. [30], food-insecure children are at risk of being malnourished because of the consumption of improper or less food. Trade-offs of food insecurity used by households, such as consuming less food and monotonous diets, may also negatively affect the nutritional status of children [29]. The finding in this review is also supported by international literature, for example, a study conducted by McQuade et al. [93] in Tanzania validated that children’s nutritional status is profoundly affected by seasonal household food insecurity, which ultimately causes malnutrition.

Similarly, Agho et al. [94] identified a positive correlation between household food insecurity and childhood malnutrition. Agho et al. [94] stated that household food insecurity influences the dietary intake of children, thus resulting in undernutrition. This finding is supported by several scholars globally, who identified household food insecurity as the primary contributor to childhood malnutrition [95,96,97,98].

#### 4.4.2. Household Income

According to Shinsugi et al. [99], low household income influences the household consumption pattern, which further affects the household’s nutritional status, causing malnutrition in children. Similarly, findings from this review reveal that low household income is a contributing factor that influences the nutritional status of children, which consequently causes malnutrition. Studies from this review that identified low household income as a factor that influences the nutritional status of children also reported a high prevalence of child malnutrition. This finding is also in line with the national survey in South Africa, which indicated that an estimated 2.5 million children under the age of five years are found in households that live below the food poverty line [13].

A low household income encourages households to purchase foods that are cheap, thus with poor nutritional value. As a result, the nutritional status of children is negatively affected [29]. Despite the provision of social grants to households with low income, a study conducted by McLaren et al. [35] revealed that children from households that receive social grants are stunted compared to children from households that do not receive social grants. Similarly, Otterbach and Rogan [7] highlight that, despite the vital role played by social grants in reducing food deprivation, households from poor settings still consume food that is poor in nutrition. Therefore, the social grant system alone is not adequate to enhance household income. A systematic review conducted in developing countries by Black et al. [100] elucidates that low household income is one of the drivers of child malnutrition. A study conducted by Asim and Nawaz [101] also agrees that household income correlates positively with child malnutrition.

#### 4.4.3. Caregiver’s Level of Education

Findings from this review suggest that the caregiver’s level of education is a factor that influences the nutritional status of children. Nguyen et al. [102] show that the low educational level of a caregiver influences the child nutritional status by affecting the household income, which profoundly influences food consumed in a household. This assertion is also supported by Ji et al. [103], who stated that a caregiver’s level of education lowers household income. Caregivers with a low educational level have a small chance of getting well-paying jobs. Phooko-Rabodiba et al. [104] found that children with caregivers who had a primary level of education were wasting and at risk of being underweight. Phooko-Rabodiba et al. [104] further elucidated that education enhances awareness and knowledge of health issues, hygiene practices, and household income. Therefore, caregivers with a low educational level may lack an understanding of hygiene practices and other health issues. According to Fonyuy and Jocelyne [105], the level of education influences the caregiver’s knowledge of feeding practices, which have a significant influence on the nutritional status of the child. Various studies from this review identified the caregiver’s nutrition education as a factor that influences a child’s nutritional status. This finding suggests that a caregiver’s knowledge of nutrition is vital for a child’s health status. Fonyuy and Jocelyne [105] stated that the lack of knowledge of a balanced diet and type of nutrients needed by the body is a critical factor that influences the child’s nutritional status and consequently malnutrition. This finding is supported by Onyeneke et al. [106], who showed that a caregiver’s knowledge of nutrition is critical in a child’s survival and development. Chege and Kuria [107] validated that caregivers who have a low educational level are more likely to have inferior nutritional knowledge, which significantly leads to child malnutrition. Sinha et al. [108] also validated that children with caregivers that were well informed about nutrition were healthier than those who had caregivers who were not well informed. Therefore, enhancing nutrition education for caregivers can act as a tool among others to fight against all forms of malnutrition.

#### 4.4.4. Household Unemployment

Results from this review show that a considerable proportion of the reviewed studies identified unemployment as a factor that influences the nutritional status of children. The high levels of unemployment reported by the reviewed studies could be attributed to the majority of predominantly rural studies. Rural areas in South Africa are still classified as areas with a high level of unemployment [109]. Phooko [51] revealed that women or child caregivers in rural areas are not allowed to work, but they must take care of the household, while men go to urban areas to work. Unemployment affects the household income and purchasing power (food), which, in turn, influences the nutritional status of children [110].

In contrast, Page et al. [111] argue that maternal employment status has no significant influence on a child’s nutritional status, because when the caregiver is unemployed, they have adequate time to provide care to their children. Rashad and Sharaf [112] agree that despite the contribution made by maternal employment status to household income, there is a positive correlation between maternal employment and child malnutrition. This finding is also supported by Brauner-Otto et al. [113], who classified maternal employment as a contributing factor that worsens the health and nutritional status of children. In contrast, Manzione et al. [114] stated that formal maternal work conditions are linked with enhanced nutritional and health status of the child. In general, in households with no one employed, the child’s nutritional status is said to be low, and the child is more vulnerable to malnutrition and other health implications [115].

#### 4.4.5. The Child’s Dietary Intake

Dietary intake has a vital role in a child’s growth and development, especially for the first 1000 days of a child’s life [116]. Perez-Escamilla, Bermudez [91] clarified that an inadequate dietary intake might deprive the child’s growth and development, making the child more vulnerable to diseases and illnesses that may ultimately result in malnutrition. Findings from this review show that dietary intake has a significant influence on child nutritional status, as reported by the reviewed studies. Nasreddine et al. [117] associated inadequate dietary intake with the consumption of foods with deficient essential nutrients and vitamins, such as iron, calcium, zinc, folic acid, vitamin A, and vitamin B12. The consumption of foods that lack essential nutrients and vitamins often results in protein-energy malnutrition (PEM), which is reflected by kwashiorkor, marasmus, and kwashiorkor-marasmus. Nasreddine et al. [117] further state that inadequate dietary intake is associated with all types of malnutrition. Boadu et al. [118] revealed that inadequate dietary intake by children results in impaired child growth and delayed cognitive development, and thus affects poor nutritional status. Bustan et al. [119] stated that low birth weight contributes significantly to stunting. Globally, researchers identify low birth weight as a driver of childhood malnutrition. For example, Trivedi et al. [120] classified low birth weight as a predictor for childhood mortality, morbidity, and malnutrition. Similarly, Meshram et al. [121] found that children with a low birth weight of less than 2500 g have a higher chance to be underweight compared to those with an average birth weight (>2500 g). Lestari et al. [122] also found a strong association between low birth weight and stunting in children under the age of five years. Comparing the nutritional status of developing countries and eating disorders of developed countries, the nutritional status of developing countries is conditioned by several factors, such poor economic growth, poverty, and frequency of diseases [117]. However, the nutritional status of developed countries is not highly affected by poverty and economic growth, but rather by eating disorders [123]. The most common eating disorders in developed countries include anorexia nervosa, bulimia nervosa, and binge-eating disorder. Eating disorders can result in severe weight loss or overweightness and obesity [123].

#### 4.4.6. Child Illness

Frequent and prolonged child illness can cause loss of appetite, absorption, metabolic disorder, and behavioral changes, which can subsequently affect the nutritional status of a child [116]. On the other hand, poor nutritional status may predispose children to illness or prolong the duration of recovery [124]. Findings from this review also identify child illness as a factor that influences the nutritional status of children. This finding is also supported by several scholars, who show that diseases such as malaria, diarrhea, vomiting, and fever negatively affect the nutritional status of children under the age of five years [124,125,126]. At the same time, malnutrition lowers the body’s capability to fight against infections by undermining the functioning of the immune response mechanism [127].

#### 4.4.7. Consumption of a Monotonous Diet

The consumption of a monotonous diet has been found in this review as a factor that affects the nutritional status of children under five years in South Africa. The consumption of a monotonous diet entails eating food that is not diverse in nutrition and may lack other essential nutrients, for example, vitamins and minerals [128]. The consumption of a monotonous diet may deprive the adequate intake of essential vitamins and minerals, which can affect the nutritional status of children [49]. This finding concurs with the study conducted in Tanzania by Blakstad et al. [129], who found that children and women who consumed a monotonous diet were malnourished compared to those who consumed a diverse food diet. Similarly, Shakya [130] showed that children who mainly consumed monotonous food, comprised primarily of carbohydrates, were nutritionally deprived. Therefore, the nutritional deprivation due to the consumption of a monotonous diet may contribute to child malnutrition.

#### 4.4.8. Poor Access to Water and Sanitation

Statistics South Africa [131] reported that an estimated 17.20% of households in South Africa still have poor access to water and sanitation. Poor access to water and sanitation has a negative impact on an individual’s health status. The finding from this review is in agreement with the finding that poor access to water and sanitation is a factor that contributes to the nutritional status of children under five years of age. Researchers all over the world also identify poor access to water and sanitation as the main factor that influences the nutritional status of children, and thus contributes to malnutrition [132,133]. Singh et al. [134] indicated that poor access to clean water often leads to child malnutrition, since the consumption of unpurified water can lead to diarrhea and other waterborne infections. This finding is also validated by Ravindranath et al. [24].

#### 4.4.9. Poor Weaning Practices

Poor weaning practices are also found to be a significant factor that influences the nutritional status of children. According to Akombi et al. [5], poor weaning practices, such as replacing breastfeeding with sucrose feeding, can deprive the child of necessary nutrients. Several scholars also identify poor weaning practices as a factor that contributes to child malnutrition [121]. The WHO recommends that children should be exclusively breastfed for at least six months, followed by the introduction of complimentary nutritious foods [135]. Therefore, correct weaning practices are necessary to enhance the nutritional status of a child. A study conducted by Syeda et al. [136] shows that children who are poorly weaned and not breastfeed are underweight and stunted.

#### 4.4.10. Gender and Age of the Child

The gender and age of the child may be critical factors that influence the nutritional status of a child. The nutrients required for boys and girls are different; this poses endless debates on the nutritional status of children in terms of their gender. However, several studies reveal that boys are more likely to suffer from malnutrition. The finding from this review also shows that the gender of the child is a critical factor that influences the nutritional status of the child, where boys are more likely to be malnourished compared to girls. There is no clear justification for this. However, Olack et al. [137] stated that boys are more likely to be malnourished because they are more influenced by environmental stress, which largely contributes to malnutrition. Schoenbuchner [135] also substantiated that a boy child is highly subjected to undernutrition.

Contrary to this, Afolabi et al. [138] showed that girls were 1.33 times more likely to be malnourished compared to boys. Besides gender, the age of the child has also been identified as a factor that can influence the nutritional status of a child. The Mother and Child Nutrition [139] alluded that children under the age of five years are highly vulnerable to malnutrition because their bodies require sufficient nutrients for growth and development, and they are easily affected by diseases that can ultimately cause malnutrition. Kassie and Workie [140] noted a high prevalence of child malnutrition in children that are below the age of five years. This observation is supported by several scholars across the globe [141].

Similarly, the assertion that the age of the child may influence the nutritional status of the child is also supported by the findings from this review. The justification for the high risk of children under five years to suffer from malnutrition may be due to poor weaning practices that may be undertaken [64]. This situation may consequently limit children from having full access to a sufficient amount of solid food with nutrients. Last but not least, this review also identified the age of the caregiver as a factor that influences the nutritional status of the child. The literature shows that children who have caregivers that are older (greater than 50 years of age) may be highly vulnerable to malnutrition because of the high levels of illiteracy among old-age people.

## 5. Limitations

Cross-sectional studies mainly dominated this review paper. This implies that this paper relied on data that was collected at a single snapshot point in time, which limited the review to identify trends of child malnutrition status in South Africa over time. As much as cross-sectional studies are efficient, inexpensive, and easy to be carried out, they can allow the researcher to be biased with the representation of results, which may affect the secondary studies [58]. On the other hand, longitudinal studies can, over time, show development, trends, and changes [59]. Therefore, the findings of this review may not reflect the accurate representation of the factors that affect nutritional status of children in South Africa.

## 6. Conclusions

This review identified several factors that affect the nutritional status of children in South Africa. These factors can be clustered into three categories, which are related to the household, child caregiver, and those that relate to the child. Factors that relate to household include food insecurity, income, unemployment, and poor access to water and sanitation, while factors that relate to the child caregiver include the caregiver’s age, level of education, and poor caregiver’s nutrition education. Factors that relate to the child include low birth weight, inadequate dietary intake, child illness, consumption of a monotonous diet, poor weaning practices, gender, and age of the child. This review further recognizes the influx of rural to informal urban migration. Yet, most research and efforts to eradicate malnutrition in South Africa are primarily rural-based or concentrated on formal urban settlements. These are critical factors to be considered in efforts to enhance the nutritional status of children in South Africa. This review argues that further research is necessary for understanding the dynamics of child malnutrition in South Africa, and should not be geographically limited to rural areas. Equally so, research is needed on poor urban settings, such as informal settlements. Therefore, in light of the findings of this review, policymakers and government practitioners need to develop strategies that are moulded by factors that affect the nutrition status of children.

Further research needs to evaluate strategies that would enhance the nutrition security of households. Findings from this review indicate that the causes of malnutrition are multifactorial, hence the need for comprehensive interventions from both the private sector and government institutions. Institutions such as the Department of Agriculture, Land Reform and Rural Development, Department of Health, and Department of Social Development all need to play a coordinating role in combating child malnutrition. This approach can be enhanced through the provision of adequate nutrition education to households, caregivers, and pregnant women. Furthermore, there must be a provision of nutritious food parcels to vulnerable households. The government should also invest in enhancing environmental health by providing adequate basic services, such as water and sanitation, to communities in South Africa.

## Figures and Tables

**Figure 1 ijerph-17-07973-f001:**
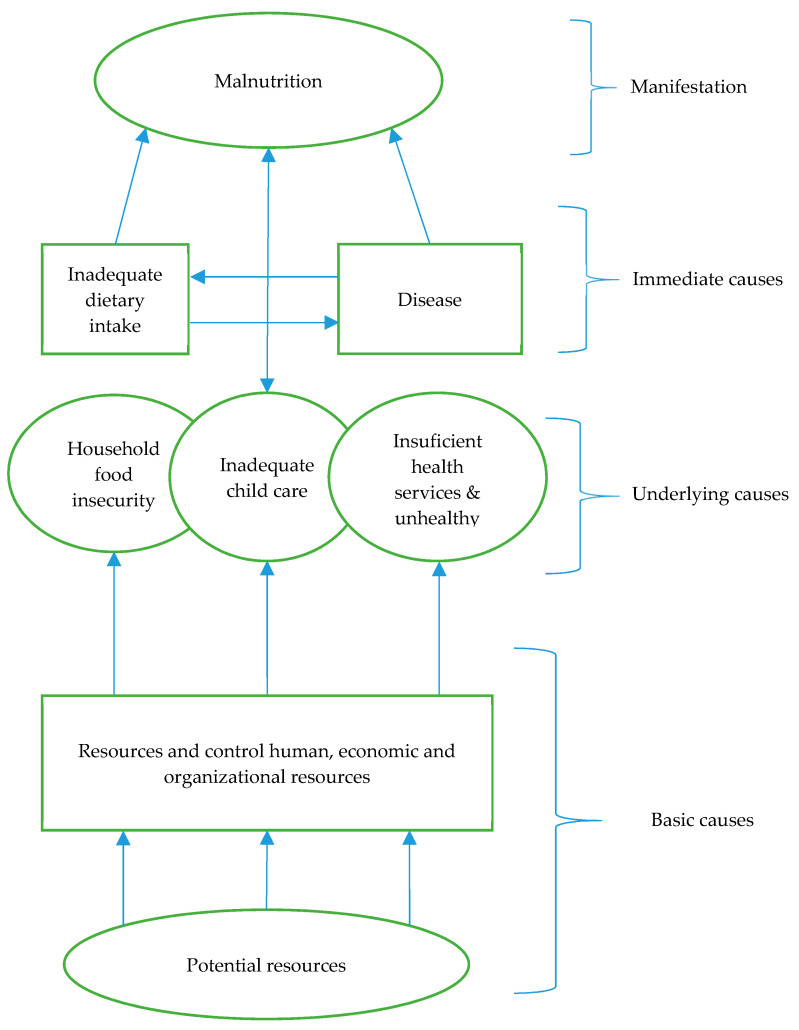
United Nations International Children’s Emergency Fund’s (UNICEF) conceptual framework of the causes of child malnutrition. Source: Adapted from Ravindranath et al. [24].

**Figure 2 ijerph-17-07973-f002:**
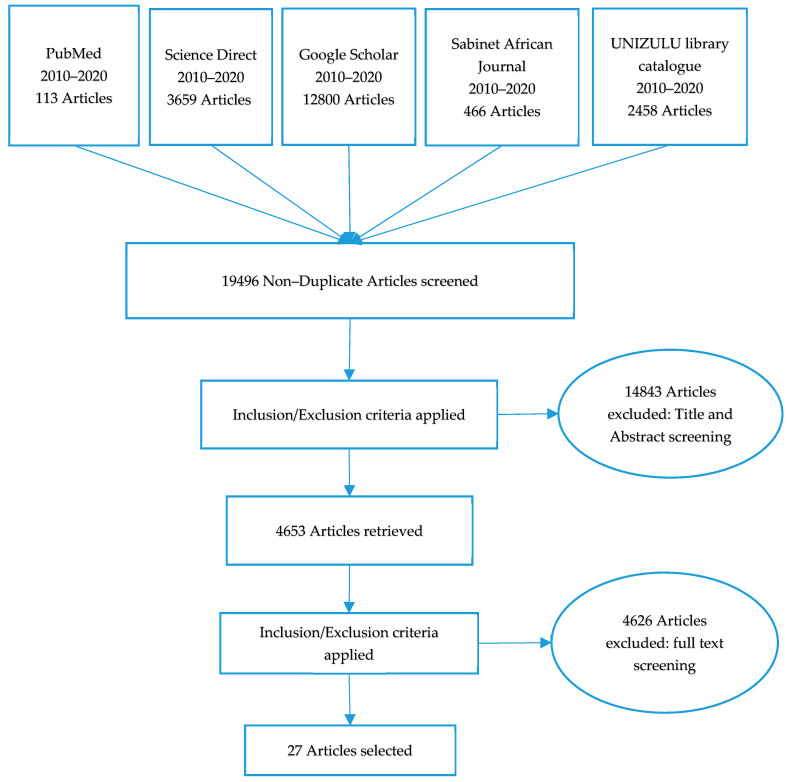
Flow chart of search strategy result.

**Figure 3 ijerph-17-07973-f003:**
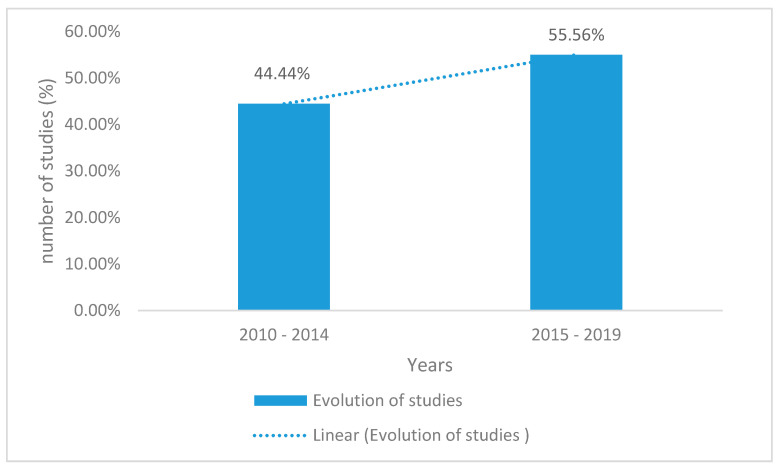
Evolution of the number of studies on child nutritional status in South Africa from the selected studies.

**Figure 4 ijerph-17-07973-f004:**
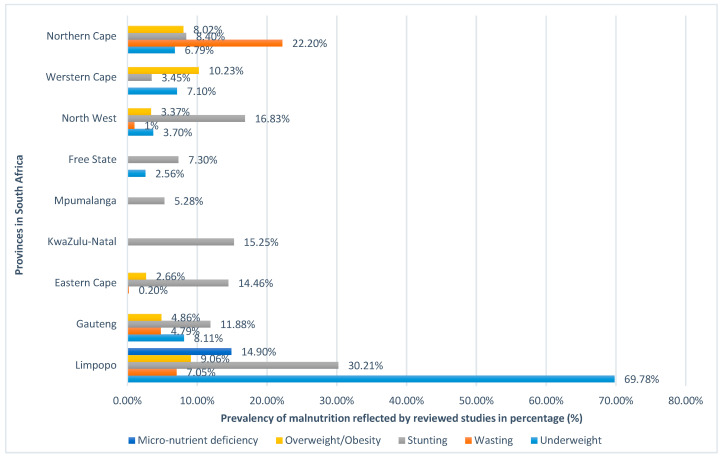
The prevalence of child malnutrition in South Africa as depicted by the selected studies.

**Figure 5 ijerph-17-07973-f005:**
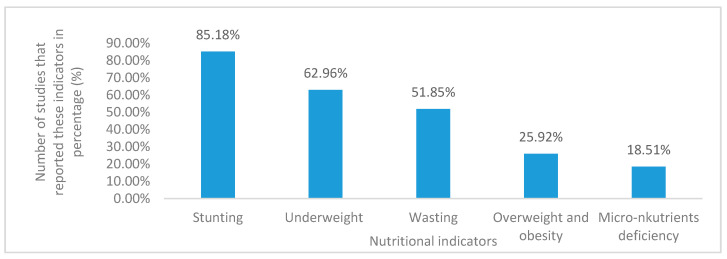
Nutritional status indicators of children in South Africa from selected studies.

**Figure 6 ijerph-17-07973-f006:**
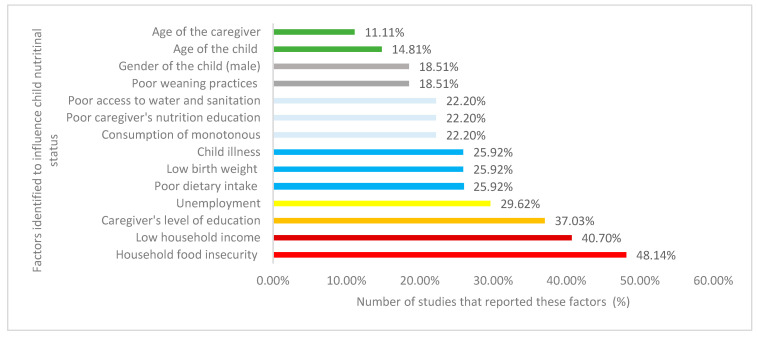
Factors influencing the nutritional status of children under five years of age in South Africa reported by the selected studies.

**Table 1 ijerph-17-07973-t001:** Criteria for inclusion and exclusion of literature sources selected for review.

Criteria for Including Literature Sources	Criteria for Excluding Literature Sources
Text documented in English	Text documented in languages aside from English
Focus is on factors that affect nutritional status	Focus is not on factors that affect nutritional status
Addresses at least one of the causes of child malnutrition identified by UNICEF’s theoretical framework of causes of child malnutrition	Addresses none of the causes of child malnutrition identified by UNICEF’s theoretical framework of causes of child malnutrition
Studies that provide detailed pertinent text needed for the review Peer-reviewed studies and articles Studies published from 2010 onwards	The text lacks pertinent details needed for review Studies and articles that are not peer-reviewed Studies published before 2010

**Table 2 ijerph-17-07973-t002:** The geographical distribution of the reviewed studies by province.

Name of Province	Frequency	Percentage (%)
Limpopo	11	40.74
Gauteng	8	29.62
Eastern Cape	6	22.22
KwaZulu-Natal	6	22.22
Mpumalanga	4	14.81
Free State	3	11.11
North West	3	11.11
Western Cape	4	14.81
Northern Cape	3	11.11
Total	48	100

**Table 3 ijerph-17-07973-t003:** A summary of the factors that influenced the nutritional status of children under five years of age in South Africa using data from the selected studies.

Location Classification (Rural/Urban)	Nutritional Indicator/s	Factors Influencing the Nutritional Status	Reference (s)/Sources
Rural areas	Stunting, underweightness, wasting	Poor access to water and sanitation, unemployment, child illness, and household food insecurity.	Schoeman et al. [28]
Rural & peri-urban	Underweightness	Nutrition insecurity, unemployment, food insecurity, monotonous diet, and inadequate dietary intake.	Ntila et al. [29]
Urban, peri-urban, informal settlement	Stunting, underweightness, wasting, overweightness, and obesity	Low birth weight, child’s gender, child’s age, age of the caregiver, unemployment, low education level, attending pre-school, low household income, and poor weaning practices.	Madiba et al. [30]
Rural	Stunting	Short maternal weight, low socioeconomic status, food insecurity, child illness, and inadequate dietary intake.	MAL-ED Network Investigators [31]
Urban	Wasting	The use of tobacco during pregnancy, unwanted pregnancy, and maternal height.	Slemming [25]
Rural	Stunting, underweightness, wasting	Unemployment, low income, poverty, illiteracy, and lack of access to adequate clean water.	Mushaphi et al. [32]
Rural	Stunting, underweightness, wasting, overweightness, and obesity	Child’s HIV status, low birth weight, caregiver’s age, and area of residence.	Kimani-Murage [33]
Rural, urban	Stunting	Low household level, poor access to water and sanitation, low quality of food, illiterate, food insecurity, and monotonous diet.	Otterbach and Rogan [7]
Urban, Rural, Urban informal settlement	Stunting, underweightness, wasting, overweightness and obesity, and micronutrient deficiency	Low birth weight, inadequate dietary intake, low household income, food insecurity, unemployment, child age, low maternal weight during pregnancy, household size, poor nutritional education, monotonous diet, child illness, type of house, and maternal BMI.	De Lange [34]
Urban	Stunting, wasting, overweightness, and obesity	Low birth weight and food insecurity.	McLaren et al. [35]
Urban	Stunting, underweightness, wasting, overweightness, and obesity	Household food insecurity, child illness, attending crèche, unemployment, illiterate, gender, and poor feeding practices.	Mahlangu and Chelule [36]
Urban, informal urban settlement	Underweightness	Low birth weight, caregiver’s inadequate nutrition education, hygiene education, age of the child, inadequate toilet facilities, and household situated in informal settlements.	le Roux et al. [37]
Rural	Undernutrition, wasting, stunting	Household food insecurity, socioeconomic status, household size, food distribution, caregiver illiterate, and unemployment.	Mandiwana et al. [38]
Rural	Stunting, underweightness, wasting, overweightness and obesity, micronutrient deficiency	Unemployment, household food insecurity, poor caregiver nutritional knowledge, inadequate dietary intake, and caregiver illiteracy.	Mushaphi [39]
Peri-urban	Underweightness, overweightness/obesity	Poor caregiver nutritional knowledge, low household income, caregiver illiteracy, unemployment, and inadequate dietary intake.	Mabweazara et al. [40]
Rural	Wasting, stunting, underweightness, micronutrient deficiency	Child gender, household food insecurity, and food feeding practices	Motadi et al. [41]
Peri-urban	Stunting	Low birth weight, maternal height, child gender, and poor maternal nutrition during pregnancy.	Matsungo et al. [42]
Rural	Undernutrition, stunting	Household size, low household income, household food insecurity, poor household infrastructure, child illness, and poor access to water and sanitation.	Schoeman et al. [43]
Rural	Stunting, underweightness, overweight	Gender of the child, no regular source of income, child illness, caregiver’s illiteracy, household food insecurity, and mother’s perception about child growth.	Lesiapeto et al. [44]
Rural	Stunting	Gender of the child, household food insecurity, poor access to piped water, distance to a health facility, socioeconomic status, and caregiver illiteracy.	Dukhi et al. [45]
Urban	Stunting	Care-giver illiteracy, home environmental factors, and asset index.	Casale et al. [46]
Informal settlement	Underweightness, stunting	Inadequate dietary intake, household food insecurity, poor dietary diversity, low household income, and caregiver illiteracy.	Selepe [47]
Rural	Stunting, underweightness, wasting	Low purchasing power, poor caregiver nutritional knowledge, unemployment, inadequate dietary intake, caregiver illiteracy, and poor feeding practice.	Kekana [48]
Urban	Stunting, underweightness, wasting	Low levels of physical activities, micro-nutrient deficiency, and inadequate dietary intake.	Nyati et al. [49]
Rural	Underweightness	Low birth weight, child’s history of malnutrition, poor sanitation, expensive formula feed, and child HIV status.	Koetaan et al. [50]
Rural	Stunting, underweightness, wasting	Unemployment, low household income, household food insecurity, and consumption of monotonous diet	Phooko [51]

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
