# Peer review of "A Review of Selected Studies on the Factors Associated with the Nutrition Status of Children Under the Age of Five Years in South Africa"

_ijerph, 2020, doi:10.3390/ijerph17217973_

Round 1

Reviewer 1 Report

Introduction - include some information on UNICEF's theoretical framework of causes of child malnutrition and link to paper's objective.

line 43 - "nutrition transition" - describe what is specifically happening in south Africa.

Methods - prisma is the standard for reviews so use it or explain why not using it.

Line 122 - exclusion chart does not mention review articles

Line 148 - figure 2 is not clear at inclusion/exclusion criteria applied step. Unclear of difference between 53 articles excluded during data extraction and 2015 excluded at full text.

Line 155 - This is not a full sentence - Studies with data that was not clear or contradicting, information obtained through reading the discussions articles.

Results - be consistent in number of decimal places used throughout

Line 166 - figure 3 adds no value and can be deleted.

Line 181 - figure 4 adds no value and can be deleted.

line 183-191 - does not make sense as studies that did not focus on nutritional status of children under 5 in south Africa should not be included in this review according to expressed criteria.

line 193-208 - this could be summarized better and refer the reader to the chart for details. When multiple studies occurred in the same area it is unclear how you came up with the prevalence, perhaps a range would be more appropriate.

line 218 - unclear if percentage or number is shown in figure 7.

line 236 - again is it number or percentage in figure 8.

Discussion - start off with stating the most important findings related to the objective of the review - review the factors influencing the
99 nutritional status of children under the age of five (5) years in South Africa - and discuss them and the implications of this for efforts to reduce malnutrition.

Overall too long, focus the information as per above.

line 244-255 - this belongs in limitations after conclusion.

Author Response

.

Reviewer 2 Report

The paper reviews the factors influencing the nutritional status of children under the age of five (5) years in South Africa in order to contribute to generating information that could be useful in formulating policies that can enhance the nutrition security among children in South Africa.

The manuscript is characterized by good scientific quality and presentation and it could be published after the authors have undertaken the proposed changes. There are, also, few typos, syntax, and grammar issues (typos, wrong word sequence, wrong tense use, missing hyphens, missing articles & commas).

  1. The introduction lacks coherence. It consists of scattered pieces of information that are reproduced again and again. Rewriting is required.
  2. Emphasize important pieces of information and try not to repeat them. The text should be on point.
  3. In my opinion, the health effects of malnutrition should be mentioned and the health symptoms of malnutrition should be analyzed in more detail, (with the corresponding references).
  4. The discussion lacks coherence in the topics. The paragraphs should have a clear topic and the information's on the same subject should be in a single paragraph e.g. a paragraph about the Stunting.
  5. It would be interesting to compare the nutrition status of the developing countries with the eating disorders of the developed countries and to report the diseases that accompany these disorders
  6. The interventions that can be made to eradicate all forms of malnutrition are not clearly stated.
  7. Replace “per cent” with one-word percent.
  8. The caregiver is one word without “-“
  9. Put “the” in front of “5 years of age”.
  10. Lines 412-424and 482-493 should have been a single paragraph. Scattered information without coherence.

Author Response

.

Reviewer 3 Report

Abstract

The abstract provides a relatively good summary of the study. One of the key findings and from this study was the lack of studies in urban areas. This needs to be highlighted in the abstract.

Line 10: Delete ‘the child’s’ replace with child

Line 15: Delete ‘South African’

Line 18: include key words used for the search

Line 27: Delete ‘were’ after articles

Introduction

The introduction contains a lot of information some of which is not relevant to the study. There is also a lot of repetition. The authors should shorten the introduction so that it highlights the need for the study. The aim of the study is also not well articulated at the end of the introduction.

The first and second paragraphs can be summarized into one/two concise paragraphs

Line 34 Delete ‘Child malnutrition continues to be an alarming global burden [1].’  This is highlighted in the second sentence

Line 37 Replace developing with low- and middle-income countries.

Consider merging summarizing line 37 and 38 into one sentence.

Line 38: Delete ‘More than one-third of all child mortality globally is due to malnutrition, although it is not recorded as the direct cause and one in four children in Africa is malnourished [5]. ‘ The previous statements highlight this.

Line 40-47: Can be summarized further. The key point is that despite major economic advancements post-apartheid, malnutrition remains a major public health problem, especially in children from poverty-stricken households. Give stats for children living in poverty in South Africa.

Line 50: Delete the 77 before ‘77and malnourished

Line 52-54: Delete “Under-nutrition is reflected by stunted growth, wasting and underweight. In contrast, over-nutrition is reflected by over-weight and obesity and micro-nutrients deficiency is indicated by the lack of essential vitamins and minerals [12]. “ We already know the types of malnutrition, focus on the statistics for each condition

Line 61-64: Highlights the negative impact of malnutrition which the authors have already discussed at the beginning of the introduction. Either delete these sentences or find a way of summarizing and incorporating them at the beginning of the introduction.

Line 65-70: Repetition- highlights the economic impact of malnutrition in more detail. This has already been discussed at the beginning of the introduction. Either delete or move to the beginning where you talk about the economic impact of malnutrition. Create a more comprehensive summary.

Line 73: Replace ‘comprehended’ with ‘understood’

Line 72-74: The authors keep shifting between poverty and malnutrition. This is a bit confusing. For example, line 72 talks about the causes of malnutrition while line 73 talks about a geographical shift in poverty. I think the main point here is not poverty but rather the higher prevalence of malnutrition in poverty stricken urban areas, hence the need to understand the causes of malnutrition?

Line 75-80: The inclusion of developed countries does not add to this paragraph. Focus on causes of malnutrition in low and middle income countries.

Line 82-84: Household income and hygiene and sanitation appear twice  

“Factors such as household food insecurity, low household income, household income, poor sanitation, lack of access to adequate clean water, insufficient health facilities, low educational level distance to health facilities, hygiene and poor sanitation[20-23].

Line 88-89: Misplaced. Doesn’t add to the paragraph. Consider deleting

Line 90: Delete ‘Besides’

Line 90-100: The aim of the study is not well defined. This paragraph can be further summarized so that the aim is clear.

Line 99-100: Repetition and part of this statement belongs to the methods section

Methods

The methods section is relatively clear but more work needs to be done on it.

Line 104-105: Not clear. Representative of what? Were the authors conducting the studies themselves?

Replace ‘for carrying out literature studies’ with ‘when selecting studies.’   

Lines 108-109 and 109-110: Repetition? What is the difference between “determined factors that influence the nutritional status of children under five (5) years” and  “Studies that evidenced factors that contribute to malnutrition in children”?

Line 111: What do you mean by with “no provincial boundaries”

Line 112: Delete ‘the’ “papers published in the international scientific”

Line 113 to 114: The authors state they only considered ‘high quality articles’ yet their description of quality is limited to their inclusion criteria. Did they consider other factors such as sample size of the studies, adjustment for bias given that studies included were cross-sectional and longitudinal studies to assess the quality of the studies?

The quality of the image of the conceptual framework can be improved. Some of the arrows are not straight

In the flow chart what is the difference between the 2015 full text articles that were excluded and 53 articles excluded during extraction? Perhaps they can go in one circle

Line 155-156: This sentence looks incomplete.

“Studies with data that was not clear or contradicting, information obtained through reading the discussions of the articles.”

Results

The authors should present more descriptive information. I would like to see a summary of:

  1. Number of children recruited (range) in the studies just to have a feel of how large these studies were.
  2. Duration of follow up for the longitudinal studies
  3. A summary of the number of studies carried out in rural, urban and urban informal settlements. This can be presented in the main text.

A description of the number of studies which provided information on Wasting, stunting, overweight and micronutrient deficiencies will also be helpful. The authors should review journal guidelines on how to present percentages.

Line 157-161: Delete. You don’t need to provide a description of the result section.

Line 163-164: You don’t need to tell the readers what cross-sectional and longitudinal studies are. Delete “where data was collected at one point in time “and “where data was collected over sometime”

Figure 3 does not add to the description of the number of articles included. Consider deleting.

Discussion

The discussion needs more work. There is a lot of information, some of which is not relevant to the aim of the study. The authors should try and make it more focused.

Start with the aim of the study and what you found out then move to study designs. Paragraph 2 should be the first paragraph as it provides a good summary of the study findings.

Information on the study designs is not very clear. The authors should highlight the strengths and limitations of each study design in relation to the study findings. Rather than providing descriptions of longitudinal and cross-sectional studies.

Line 301 to 334: Consider deleting. Issues such as prevalence should not be on the fore front because the aim of the study was not to assess the prevalence of malnutrition, but rather the types of risk factors in different locations. The effects of undernutrition, specifically wasting and stunting should be excluded.

Line 342-351: Talks about obesity and overweight, but the information provided is not focused and some of it is not relevant. The paragraph should start with risk factors of obesity and overweight identified by the authors. They should then proceed to compare studies with similar findings and then move on to provide possible explanations for the risk factors

Why were most of the studies conducted in Limpompo?

Line 537-546: I feel this do not add to the review and should be deleted. The conclusion should start from line 547.

Line 557-559 Consider deleting. You have already highlighted the negative impact of malnutrition in the introduction.

Line 565-568: Rewrite to highlight that findings from the review suggest that the causes of malnutrition are multifactorial hence the need for comprehensive interventions

Author Response

.

Round 2

Reviewer 1 Report

Figure 2 - numbers don't add up

Figure 3 does not make sense as 100% of the studies reviewed should be on this topic

lines 203-219 could be better summarized to highlight most important points related to figure

Figure 4 - unclear how percentages were arrived at. Was an average taken or should a range be included?

Line 231-246 could be better summarized to highlight most important points related to figure

Some previous reviewer comments not well addressed.

Author Response

.

Reviewer 3 Report

The authors have improved the manuscript significantly but there are still a few things that need to be addressed.

Abstract

Line 21: Add ‘s’ at the end of the word survey so that it reads surveys

Methods

More on the details on how you assessed quality, a large sample size does not always translate to a high-quality study- did you assess bias and adjustment for confounding factors in the selected studies?

Lines 123-130 should be moved to the either the introduction section or the discussion.

Line 621: Should be mother instead of Mather  

 Discussion

The limitations of the study should come before the conclusion. The conclusion should be the last section of the paper.

I feel the fact that most of the studies identified in the review is not a limitation of the study but a key finding, unless you purposefully selected studies conducted in rural areas.

Perhaps you can move line 322-329 to the limitations section

Line 385 and 387: Replace United Nations International Children Emergency Fund with UNICEF

Author Response

.

Round 3

Reviewer 1 Report

The paper should be edited to address English language concerns.